# UltraG-Ray: Physics-Based Gaussian Ray Casting for Novel Ultrasound View Synthesis

**Felix Duelmer**[1,2]  🆔        FELIX.DUELMER@TUM.DE
**Jakob Klaushofer**[1]            JAKOB.KLAUSHOFER@TUM.DE
**Magdalena Wysocki**[1,2]  🆔   MAGDALENA.WYSOCKI@TUM.DE
**Nassir Navab**[1,2]  🆔         NASSIR.NAVAB@TUM.DE
**Mohammad Farid Azampour**[1,2]  🆔   MF.AZAMPOUR@TUM.DE
[1] *Chair for Computer Aided Medical Procedures (CAMP), Technical University of Munich, Germany*
[2] *Munich Center for Machine Learning (MCML), Munich, Germany*

**Editors:** Accepted for publication at MIDL 2026

## Abstract

Novel view synthesis (NVS) in ultrasound has gained attention as a technique for generating anatomically plausible views beyond the acquired frames, offering new capabilities for training clinicians or data augmentation. However, current methods struggle with complex tissue and view-dependent acoustic effects. Physics-based NVS aims to address these limitations by including the ultrasound image formation process into the simulation. Recent approaches combine a learnable implicit scene representation with an ultrasound-specific rendering module, yet a substantial gap between simulation and reality remains. In this work, we introduce UltraG-Ray, a novel ultrasound scene representation based on a learnable 3D Gaussian field, coupled to an efficient physics-based module for B-mode synthesis. We explicitly encode ultrasound-specific parameters, such as attenuation and reflection, into a Gaussian-based spatial representation and realize image synthesis within a novel ray casting scheme. In contrast to previous methods, this approach naturally captures view-dependent attenuation effects, thereby enabling the generation of physically informed B-mode images with increased realism. We compare our method to state-of-the-art and observe consistent gains in image quality metrics (up to 15% increase on MS-SSIM), demonstrating clear improvement in terms of realism of the synthesized ultrasound images.

**Keywords:** Ultrasound, Novel View Synthesis, Volumetric Representation, Gaussian Splatting, 3D Ultrasound Reconstruction

## 1. Introduction

Routine ultrasound examinations are based on 2D B-mode images (Adriaans et al., 2024), forcing clinicians to mentally reconstruct 3D anatomy from multiple slices, a process that depends heavily on spatial reasoning and experience (Kojcev et al., 2017; Krönke et al., 2022). This cognitively demanding task can be alleviated by 3D reconstruction methods, which provide spatially coherent representations that support more reliable anatomical interpretation. When probe poses or overlapping views are available, these methods combine the acquired slices into a coherent representation of the scanned region and enable retrospective generation of views beyond the physically acquired plane, referred to as novel view synthesis (NVS). NVS has been gaining importance in clinical practice, as it can support simulators for training of clinicians (Blum et al., 2013; Ehricke, 1998), assist clinicians when

reviewing data from robotic ultrasound systems (Jiang et al., 2023; Bi et al., 2024; Cao et al., 2025), and can potentially provide a training environment for emerging ultrasound world models (Yue et al., 2025; Qu et al., 2025).

A straightforward strategy for ultrasound NVS is to compound 2D frames into a 3D volume, either via freehand acquisition (Prevost et al., 2018; Wilson et al., 2025) or with robotic, optical, or electromagnetic tracking for higher accuracy (Jiang et al., 2023; Adriaans et al., 2024), and then re-slice the volume. However, ultrasound is highly anisotropic: intensity values corresponding to the same voxel but acquired from different orientations can disagree, producing inconsistent or blurred representations. Simple aggregation schemes such as mean or maximum compounding (Lasso et al., 2014) cannot resolve these orientation-dependent ambiguities and often fail to preserve the true underlying tissue structure.

View-dependent representations were introduced to model the strong dependence of ultrasound appearance on the probe position and orientation, which conventional view-independent reconstructions cannot capture. This motivation led to computational sonography (Hennersperger et al., 2015; Göbl et al., 2018) which models an ultrasound volume as a vector field in which each voxel stores an array of orientation-dependent intensity values, allowing view-specific appearance. However, this voxelized encoding of viewing directions imposes coarse angular resolution and interpolation artifacts, yielding blurred B-mode images. It also lacks physical consistency, reproducing only appearances seen during training.

Recent developments in NeRF-based methods have been proposed for ultrasound to mitigate these limitations (Wysocki et al., 2024; Dagli et al., 2024; Gaits et al., 2024; Grutman et al., 2025). These approaches use pose-annotated ultrasound images to learn an implicit continuous representation of ultrasound-specific parameters, coupled to a differentiable, ray-casting-based forward-synthesis module that models anisotropy in B-mode images. However, their sampling-based volumetric rendering inherently smooths out high-frequency structures, which still limits realism.

While the physical origin of image formation differs fundamentally between ultrasound and natural images, sampling-based volumetric integration in NeRF leads to analogous over-smoothing effects in both domains. In natural-image NVS, this limitation has been addressed by 3D Gaussian Splatting (3DGS) (Kerbl et al., 2023). By replacing implicit volumetric fields with an explicit set of learnable 3D Gaussians and a GPU-efficient differentiable rasterizer, 3DGS enables significantly sharper reconstructions and real-time rendering performance. In the medical domain, 3DGS has already been applied to endoscopic scene representation (Liu et al., 2025), novel-view synthesis in X-ray (Cai et al., 2024), and intraoperative surgical navigation (Fehrentz et al., 2025).

In ultrasound, two methods using the Gaussian representation have been introduced in this context: UltraGS (Yang et al., 2025) and UltraGauss (Eid et al., 2025). UltraGS follows the original idea of projecting Gaussians onto a two-dimensional canvas. However, the approach assumes placing a virtual camera inside the tissue, directly in front of the acquired B-mode image. This setup would require an explicit selection mechanism to determine which Gaussians fall within the thin slab of tissue corresponding to the current ultrasound slice, yet no such mechanism is specified in the original description (Yang et al., 2025). UltraGauss, on the other hand, follows a more promising direction by avoiding projection onto a distant plane and instead computing the intersection between each Gaussian and the image plane. Although this formulation is better aligned with the acquisition process, it

does not include ultrasound-specific mechanisms for handling shadowing effects or resolving ambiguities caused by reflective structures (Eid et al., 2025).

In this paper, we introduce UltraG-Ray, a novel 3D Gaussian-based method that addresses these limitations by introducing a ray casting framework embedded in a Gaussian representation that follows the physics of ultrasound image formation (compare Fig. 1). Instead of relying on plane intersections or 2D projections, we follow the direction of acoustic wave propagation and incorporate attenuation directly into the 3D Gaussian field. This allows us to model depth-dependent signal decay, shadowing from strongly absorbing or reflecting structures, and the orientation-dependent visibility changes characteristic of real B-mode imaging. By capturing these ultrasound-specific mechanisms, our approach enables anatomically consistent, view-dependent NVS with enhanced visual realism. In summary, our contributions are as follows:

- A novel, high-fidelity NVS method for ultrasound that uses a learnable 3D Gaussian representation encoding ultrasound-specific parameters such as reflection and attenuation, combined with a physics-based, ray casting forward model for view-dependent real-time B-mode generation.
- An ultrasound-specific optimization strategy to prune, duplicate, and split Gaussians to enhance scene representation.
- Two open-source, pose-annotated datasets (in-silico and ex-vivo) containing overlapping views and various viewing angles.

## 2. Basics of Gaussian Splatting

3DGS (Kerbl et al., 2023) builds upon two established ideas in computer graphics: representing scenes using anisotropic 3D Gaussians with elliptical rasterization (Zwicker et al., 2002), and compositing their contributions using an emission–absorption model as in volumetric rendering (Mildenhall et al., 2021). Each Gaussian is defined by a mean $\boldsymbol{\mu}_i \in \mathbb{R}^3$, a covariance matrix $\boldsymbol{\Sigma}_i \in \mathbb{R}^{3 \times 3}$, an opacity $\alpha_i$, and a color vector $\mathbf{c}_i \in \mathbb{R}^3$.

In practice, $\mathbf{c}_i$ is parameterized using low-order spherical harmonics (SH): for a viewing direction $\mathbf{v}$, the emitted color of Gaussian $i$ is given by

$$\mathbf{c}_i(\mathbf{v}) = \sum_{b=1}^{B} \boldsymbol{\beta}_{i,b} Y_b(\mathbf{v}), \tag{1}$$

where $Y_b$ are spherical harmonic basis functions and $\boldsymbol{\beta}_{i,b} \in \mathbb{R}^3$ are learned coefficients. The number of terms $B$ is set by the SH degree. This enables a compact, view-dependent appearance per Gaussian. To render this representation from a given viewpoint, each Gaussian must be projected onto the image plane. Let $\Pi(\cdot)$ denote the camera projection and $\mathbf{J}_i$ the Jacobian of $\Pi$ evaluated at $\boldsymbol{\mu}_i$. The projection yields a 2D Gaussian footprint with mean $\mathbf{u}_i = \Pi(\boldsymbol{\mu}_i)$ and covariance $\boldsymbol{\Sigma}_i^{2D} = \mathbf{J}_i \, \boldsymbol{\Sigma}_i \, \mathbf{J}_i^{\top}$.

For a pixel location $\mathbf{p}$, the contribution of Gaussian $i$ is determined by the Mahalanobis-weighted footprint

$$w_i(\mathbf{p}) = \exp\!\left(-\tfrac{1}{2}(\mathbf{p} - \mathbf{u}_i)^{\top} (\boldsymbol{\Sigma}_i^{2D})^{-1} (\mathbf{p} - \mathbf{u}_i)\right), \tag{2}$$

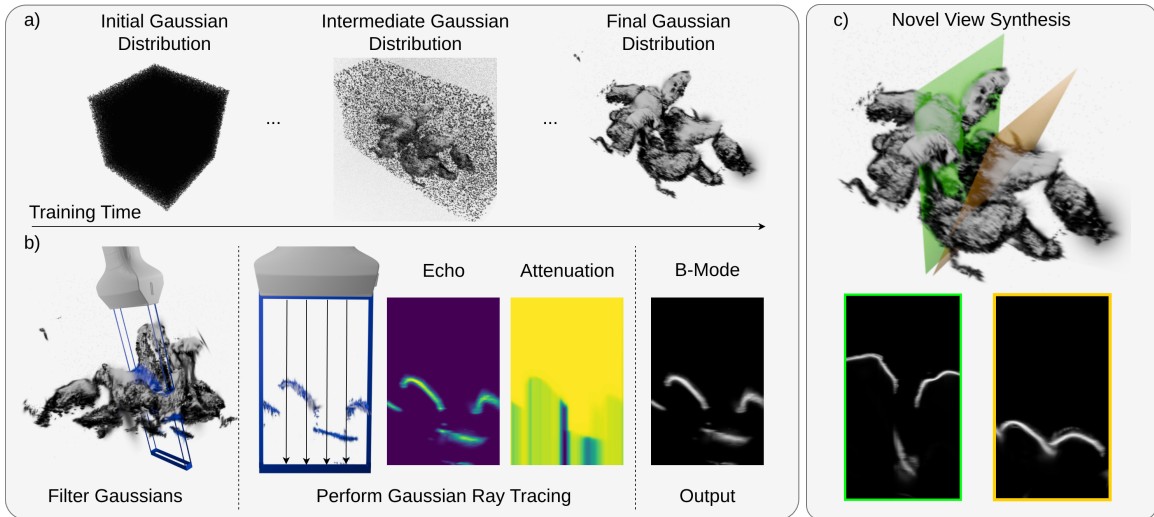

Figure 1: Overview of UltraG-Ray pipeline: a) Progressive adaptation of the learnable 3D Gaussian distribution to match the acquired data b) Image synthesis pipeline: First Gaussians are filtered based on the respective pose, second Gaussian ray-intersection is used to create the echo and transmittance maps and finally the resulting B-Mode image c) Downstream task evaluation based on NVS, where green and orange frames denote novel views

which smoothly decays with distance from the projected center. This footprint effectively modulates the effective opacity as $\alpha_i(\mathbf{p}) = \alpha_i \, w_i(\mathbf{p})$. Because multiple Gaussians can overlap along the viewing direction, their contributions must be composited in a physically meaningful order. All elements are therefore sorted by depth (front-to-back), and visibility is accumulated using transmittance. Starting with $T_0 = 1$, the transmittance after Gaussian $i$ is given by

$$T_i(\mathbf{p}) = T_{i-1}(\mathbf{p}) \left(1 - \alpha_i(\mathbf{p})\right), \tag{3}$$

which accounts for the fraction of light that remains unoccluded after passing through the first $i$ Gaussians. This recursion is equivalent to standard front-to-back alpha blending used in computer graphics. Finally, the pixel color follows the standard emission–absorption formulation

$$C(\mathbf{p}, \mathbf{v}) = \sum_{i=1}^{N} T_{i-1}(\mathbf{p}) \, \alpha_i(\mathbf{p}) \, \mathbf{c}_i(\mathbf{v}), \tag{4}$$

where each Gaussian contributes according to its (view-dependent) color, its opacity at $\mathbf{p}$, and its visibility through previously encountered Gaussians. This formulation allows the method to efficiently approximate complex view-dependent appearance while remaining fully differentiable.

## 3. Methods

### 3.1. Ultrasound Physics

An ultrasound image is typically generated by a transducer positioned to face the tissue and emit acoustic waves into it. Due to the lenses and programmable focusing, the waves concentrate in a thin, but elongated, volume in front of the transducer. Tissue parameters, such as density or the speed of sound, determine how the wave propagates through the body. Due to inhomogeneities caused by variations in these tissues, sound is reflected back to the transducer. Next to this reflection, causing a weakening of the wave, the amplitude of the wave is being attenuated due to spherical decay and relaxation processes. Upon reception of the echo at the transducer surface, the signal is processed through a pipeline where it is beamformed, envelope detected, and log-compressed (Hoskins et al., 2019).

### 3.2. Ultrasound-Adaptive Gaussian Forward Model

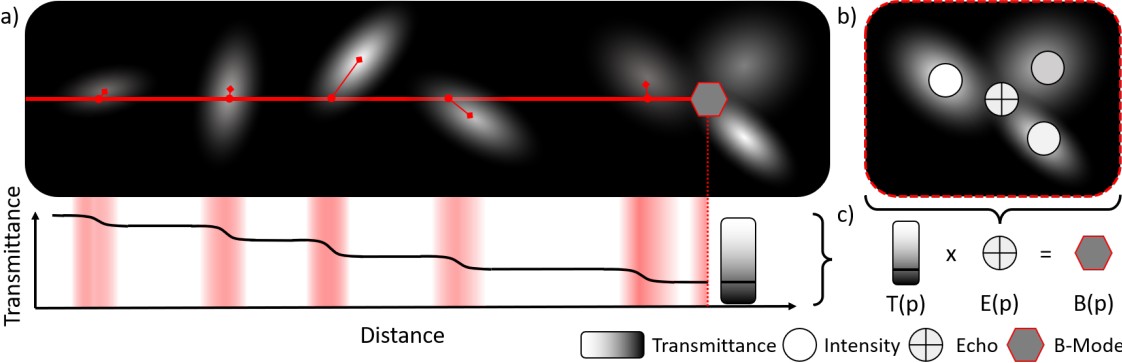

Figure 2: Depiction of the attenuation and intensity formation process: a) Transmittance accumulation, where each Gaussian gradually reduces the remaining energy along the ray. b) Computation of pixel-wise intensity contributions from view-dependent Gaussian intensity and their distance-weighted influence. c) Final pixel intensity obtained by combining accumulated attenuation with the weighted Gaussian contributions.

Acoustic attenuation differs fundamentally from light attenuation (Duelmer et al., 2025), making the standard RGB-based representation through color and opacity insufficient to capture the unique characteristics of ultrasound physics. To model ultrasound attenuation, we build on a ray casting scheme previously introduced in (Salehi et al., 2015), with important modifications enabled by the Gaussian formulation. In particular, we simplify the equations by leveraging the Gaussians' ability to approximate a point-spread function (PSF). Freed from a discretized sampling grid, speckle is represented directly with explicit Gaussians instead of as a distribution.

In UltraG-Ray, we define every 3D Gaussian by a mean $\boldsymbol{\mu}_i \in \mathbb{R}^3$, a covariance matrix $\boldsymbol{\Sigma}_i \in \mathbb{R}^{3\times3}$, a transmittance value $\tau_i$, and an intensity value $I_i$, which together parameterize the Gaussian used at rendering time. The parameters $\boldsymbol{\mu}_i$, $\tau_i$, and $I_i$ are directly optimized.

To compute $\boldsymbol{\Sigma}_i$, we parameterize each Gaussian by a scale vector $\mathbf{s}_i$ and a quaternion $\mathbf{q}_i$, from which the full covariance $\boldsymbol{\Sigma}_i$ is obtained analytically. We use this reparameterization because it ensures a stable and unconstrained optimization of anisotropic Gaussian shapes while guaranteeing that the resulting covariance matrices remain positive definite.

To ensure efficient computation, we cull Gaussians that cannot contribute to the final image. In this step, each Gaussian is projected onto the far plane, corresponding to the ultrasound imaging depth, to compute its 2D footprint. The projection follows an orthographic camera model defined by the transducer's pose and lateral width. For every potential ray endpoint, we define a small rectangular region on the far plane and test whether the projected Gaussian footprint intersects this rectangle. Only Gaussians whose projected support overlaps these rectangles are forwarded to the ray casting module. Note that the vertical extent is not set to the elevational height of the transducer but rather to a small value to approximate the 2D ultrasound imaging plane. Having restricted the set of Gaussians to those that can influence the image, we next characterize how each remaining Gaussian contributes to the intensity of a pixel. We model the B-mode pixel value as:

$$B(p) = T(p) \cdot E(p), \tag{5}$$

In this factorization, $T(\mathbf{p})$ denotes the transmission factor, which is the fraction of the emitted acoustic energy that remains available at location $\mathbf{p}$ along the scan line after cumulative attenuation on the forward transmit path. The term $E(\mathbf{p})$ denotes the local echo amplitude generated at $\mathbf{p}$ by backscattering and specular reflections, and it represents the strength of the received contribution (compare Fig. 2). To model acoustic attenuation, we approximate wave interference with a ray casting model, treating each scan line as a ray emitted from the origin on the transducer surface $\mathbf{o}$ and direction $\mathbf{d}$ given by the local surface normal (i.e., orthogonal to the transducer surface), parametrized by $r(z) = \mathbf{o} + z\mathbf{d}$. To compute the contribution of a Gaussian $G_i$ to local energy decay, we transform the ray into its canonical space following (Moenne-Loccoz et al., 2024; Yu et al., 2024). This change of variables applies the inverse anisotropic scaling and rotation that map the Gaussian in world coordinates to a standard normal distribution with identity covariance:

$$\mathbf{o}_g = S_i^{-1} R_i^\top (\mathbf{o} - \boldsymbol{\mu}_i), \qquad \mathbf{d}_g = S_i^{-1} R_i^\top \mathbf{d}, \qquad \ell_g = \|\mathbf{d}_g\| \, \ell. \tag{6}$$

Here $R_i$ and $S_i$ are the Gaussian's rotation and scale, and $\ell$ denotes the ray segment length in world space. As a result, the anisotropic Gaussian becomes isotropic in canonical space, which enables an efficient and numerically stable evaluation of the subsequent line integral:

$$\psi_i = \int_0^\ell \exp\left(-\tfrac{1}{2} \|\mathbf{o}_g + t\, \mathbf{d}_g\|^2\right) dt. \tag{7}$$

This line integral measures the overlap between the scan line and Gaussian $G_i$, and it increases when the ray passes closer to $\boldsymbol{\mu}_i$. We approximate this integral numerically using a low-order Gaussian quadrature with three evaluation points, namely, the 3-point Gauss-Legendre quadrature. Concretely, we integrate over a short interval centered at the point of closest approach between the ray and the Gaussian, where the integrand concentrates most of its mass. This yields an accurate and differentiable estimate with constant computational cost per Gaussian. We map the ray–Gaussian integral $\psi_i$ directly to a per-Gaussian

transmittance and define the total transmission along the ray as:

$$T(z) = \prod_{i=1}^{n} T_i, \text{ where } T_i = \tau_i + (1 - \tau_i)\exp(-\psi_i). \tag{8}$$

Here, $\tau_i$ is the learnable transmittance of Gaussian $G_i$, $\exp(-\psi_i)$ represents the physically motivated attenuation induced by the Gaussian, and the product over $T_i$ yields a multiplicative attenuation model along depth that is consistent with Beer–Lambert type exponential decay, used here as an effective approximation for ultrasound. This approximation is sufficient in practice because clinical ultrasound systems apply depth-dependent gain and log compression that largely compensate for global exponential decay.

For the echo signal $E(\mathbf{p})$, we compute the intensity contributed by each Gaussian based on its Mahalanobis-weighted distance to the pixel. Unlike splatting Gaussians in the 2D image domain (see Eq. 2), we follow (Eid et al., 2025) and evaluate this contribution directly in 3D using the Mahalanobis distance: $w_i(\mathbf{p}) = \exp\left(-\frac{1}{2}(\mathbf{p} - \boldsymbol{\mu}_i)^\top (\boldsymbol{\Sigma}_i)^{-1} (\mathbf{p} - \boldsymbol{\mu}_i)\right)$.

To capture view-dependent backscattering, we parameterize the per-Gaussian echo amplitude as a low-order (degree-1) spherical harmonic (SH) expansion along ray direction $\mathbf{d}$ simplified to a single channel for intensity calculation (see Section 2 and Eq. 1). To ensure smoothness in regions where few or no Gaussians contribute to a pixel, we introduce a soft coverage mechanism that modulates the rendered intensity based on the strength of the pixel's support from the Gaussian field. Let $S(\mathbf{p}) = \sum_{i=1}^{n} w_i(\mathbf{p})$ denote the total Gaussian footprint at pixel $\mathbf{p}$. We define a soft coverage factor: $\gamma(\mathbf{p}) = 1 - \exp(-S(\mathbf{p}))$, which remains close to zero when the pixel is only weakly covered by Gaussians and approaches one as the local footprint mass increases. The final echo intensity is then given by

$$E(\mathbf{p}) = (1 - \gamma(\mathbf{p}))\, I_{\text{bkg}} + \gamma(\mathbf{p})\, \frac{\sum_{i=1}^{n} I_i(\mathbf{d})\, w_i(\mathbf{p})}{S(\mathbf{p}) + \varepsilon}, \tag{9}$$

with a small $\varepsilon > 0$ for numerical stability and a constant background intensity $I_{\text{bkg}} \in \mathbb{R}$ representing echo-free regions in ultrasound, which we set to black for the whole image.

### 3.3. Initialization and Optimization Strategy

Conventional 3DGS relies on structure-from-motion (Schonberger and Frahm, 2016) to initialize Gaussians and camera poses, which is incompatible with our data due to the fundamentally different image formation in ultrasound. We therefore initialize all Gaussians randomly within the local scene and set the remaining parameters to defaults (see App. A).

Like in 3DGS, we optimize all Gaussian parameters using stochastic gradient descent. To ensure agreement between the rendered slices and the ground truth, we employ a reconstruction objective composed of an L1 loss and a SSIM loss. Additionally, we regularize the Gaussian scales to encourage non-contributing or excessively large Gaussians to shrink. The complete training loss is therefore defined as:

$$\mathcal{L} = (1 - \lambda_{\text{ssim}})\, \mathcal{L}_1 + \lambda_{\text{ssim}}\, \mathcal{L}_{\text{SSIM}} + \lambda_{\text{scale}}\, \mathbb{E}[\exp(\mathbf{S})]. \tag{10}$$

Optimizing with this loss alone is insufficient because a refinement mechanism is required to add Gaussians where detail is missing and remove those that do not contribute. Since

existing schemes are incompatible with our custom synthesis pipeline, we use a 3DGS-inspired refinement strategy (Kerbl et al., 2023) adapted to ultrasound data. After a brief warm-up, refinement is triggered every $t$ iterations and applies pruning, duplication, and splitting to maintain a compact and expressive Gaussian set. Gaussians with negligible or excessively large scales, which either fail to contribute or oversmooth local structure, are pruned.

By tracking the gradients of each Gaussian over $t$ iterations, we compute a mean importance score that determines whether a Gaussian should be modified during refinement. Gaussians whose importance exceeds a predefined threshold are either duplicated or split, depending on their scale. Small Gaussians are duplicated to increase local representational capacity, while larger Gaussians are split into two nearby Gaussians to refine structure. We cap the total number of Gaussians to balance computational cost and reconstruction fidelity. Because real ultrasound acquires a thin elevational volume rather than infinitesimally thin rays (see Sec. 3.1), we approximate this effect by perturbing all ray origins with a small out-of-plane offset drawn from a cosine-weighted distribution. This perturbation also discourages the Gaussians from collapsing into thin 2D structures tied to individual frames.

### 3.4. Implementation

We implement UltraG-Ray[1] in PyTorch on top of the CUDA-accelerated 3D Gaussian splatting library gsplat (Ye et al., 2025). Our implementation adds custom CUDA kernels for selecting relevant Gaussians per pixel and for efficiently evaluating the ray–Gaussian equations described in Sec. 3.2. All experiments are run on a single desktop workstation with an Intel Core i7-12700 CPU and an NVIDIA RTX 4070 Ti GPU. For training, we use the Adam optimizer with parameter-specific learning rates. We set the loss weights to $\lambda_{\text{ssim}} = 0.5$ and $\lambda_{\text{scale}} = 10^{-3}$ and train for 30k epochs, which we found sufficient for convergence in all experiments. To approximate the elevational beam width, we perturb all ray origins with an out-of-plane offset sampled from a cosine-weighted distribution with maximum magnitude $\delta = 2\,\text{mm}$, which is within the range of elevational slice thicknesses for clinical transducers (Scholten et al., 2023). Additional information about parameters used in training and optimization strategy can be found in Appendix A.

## 4. Experiments and Results

**Datasets**  To validate UltraG-Ray, we introduce two pose-annotated B-mode datasets. To demonstrate robustness across both strong reflectors and heterogeneous scattering, we evaluate on an in-silico spine phantom and an ex-vivo porcine muscle phantom. Data were acquired with a Siemens 12L3 probe connected to a Siemens Acuson Juniper system mounted on a KUKA LBR iiwa 14 R820 robot using a custom probe holder. Image and tracking data were recorded via the software ImFusion (ImFusion GmbH, Munich, Germany). We first applied a coarse calibration based on the probe geometry and the known offsets from the robot's last joint. This was then refined using a calibration procedure based on the BOBYQA optimization algorithm (Powell et al., 2009), implemented in the software

---

[1]Both datasets and the implementation can be found here: https://github.com/jakobkla/UltraG-Ray

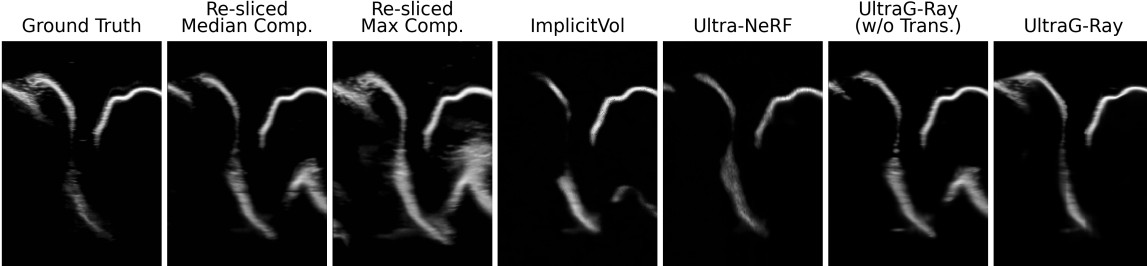

Figure 3: Qualitative comparison of UltraG-Ray and baselines on the spine phantom.

ImFusion. For a detailed hardware setup overview, we refer the reader to (Jiang et al., 2023). Both datasets comprise multiple overlapping sweeps acquired at different probe tilt angles. The porcine muscle dataset covers an extent of approximately 5 cm, yielding a volume of about $5 \times 5 \times 5$ cm$^3$. It contains three sweeps acquired at $-3°$, $0°$, and $+3°$, and we evaluate on the $-3°$ sweep. In addition, we acquire sweeps at $-5°$, $-7°$, and $-10°$ that are held out exclusively for ablation testing (see App. D). Each sweep comprises roughly 100–120 frames with associated poses. The spine phantom dataset consists of six sweeps acquired at $-20°$, $-10°$, $0°$, $+10°$, and $+20°$, with evaluation performed on an additional $+15°$ sweep. The covered volume is approximately $13 \times 5 \times 9$ cm$^3$, and each sweep provides around 350–400 frames with associated poses. GPU memory requirements vary with the number of Gaussians. Assuming 500k Gaussians, peak GPU memory is approximately 1.4 GB. End-to-end training takes $\approx 12$ min for the spine phantom and $\approx 80$ min for the porcine muscle phantom.

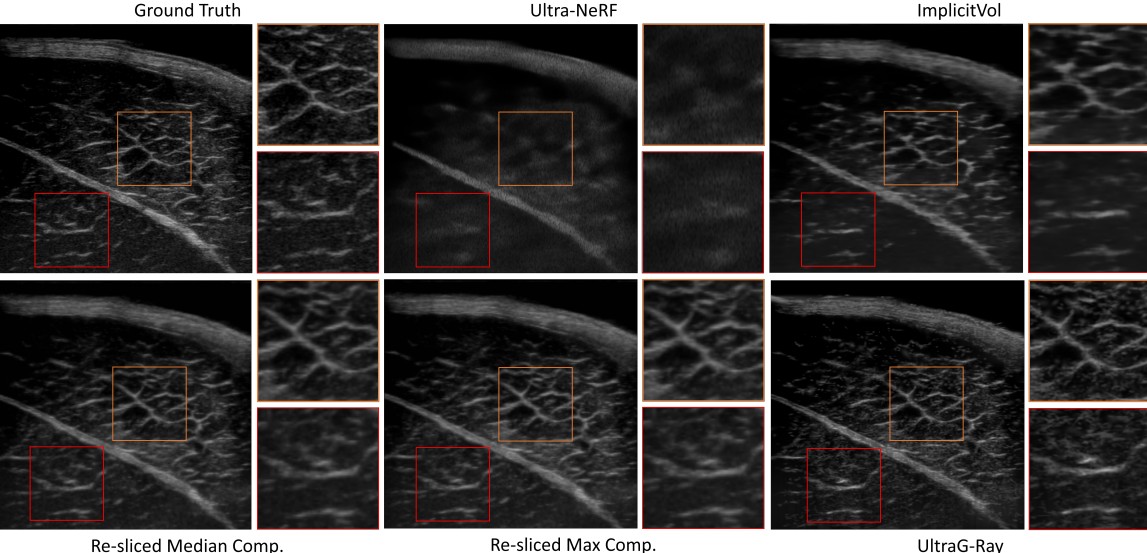

Figure 4: NVS comparison on ex vivo porcine muscle. Highlighted regions (red and orange) are enlarged as cut-outs on the right side of the respective image.

**Novel View Synthesis** We compare our method against Ultra-NeRF (Wysocki et al., 2024) and ImplicitVol (Yeung et al., 2021). ImplicitVol reconstructs a continuous 3D ultrasound volume as a learned implicit function from 2D frames, without modeling view-dependent appearance. Implementation details for both baselines are provided in App. C. In addition, we include two traditional, non-learning-based baselines based on volume compounding and re-slicing. Specifically, we compound the training sweeps into a 3D volume using either maximum or median compounding at an isotropic voxel size of 0.25 mm, and subsequently re-slice the volume with linear interpolation to obtain the corresponding image planes. Both compounding and re-slicing are performed using implementations from ImFusion. As both UltraGS (Yang et al., 2025) and UltraGauss (Eid et al., 2025) did not make their source code publicly available[2], we can not directly compare to their implementations. In particular, reproducing the original methods ourselves would risk not doing justice to the respective approaches. For quantitative evaluation, we report multi-scale structural similarity (MS-SSIM) and peak signal-to-noise ratio (PSNR). As these metrics can favor overly smooth reconstructions, we additionally include gradient-magnitude similarity (GMS) and gradient-magnitude similarity deviation (GMSD) (Xue et al., 2013), which better capture high-frequency detail and edge preservation.

Table 1: Evaluation results for NVS on both datasets. Best values are in bold. Arrows indicate whether higher ($\uparrow$) or lower ($\downarrow$) values correspond to better performance.

| Dataset | Method | PSNR $\uparrow$ | MS-SSIM $\uparrow$ | GMS $\uparrow$ | GMSD $\downarrow$ |
|---|---|---|---|---|---|
| Porcine Muscle | Re-sliced Median Comp. | 24.96±0.67 | 0.73±0.03 | 0.87±0.01 | 0.17±0.01 |
| | Re-sliced Max Comp. | 24.18±0.59 | 0.73±0.03 | 0.87±0.01 | 0.17±0.01 |
| | Ultra-NeRF | 23.60±0.54 | 0.62±0.03 | 0.81±0.02 | 0.23±0.01 |
| | ImplicitVol | 25.16±0.56 | 0.74±0.02 | 0.85±0.01 | 0.19±0.01 |
| | UltraG-Ray (w/o Trans.) | 24.31±0.76 | 0.74±0.03 | 0.86±0.01 | 0.18±0.00 |
| | UltraG-Ray (SH=0) | 25.10±0.69 | 0.76±0.02 | 0.87±0.01 | 0.17±0.01 |
| | UltraG-Ray (no OPS[3]) | 23.39±1.74 | 0.68±0.07 | 0.86±0.02 | 0.18±0.01 |
| | UltraG-Ray | **25.49**±0.60 | **0.77**±0.02 | **0.88**±0.01 | **0.16**±0.01 |
| Spine Phantom | Re-sliced Median Comp. | 25.83±4.27 | 0.93±0.05 | 0.96±0.02 | 0.13±0.05 |
| | Re-sliced Max Comp. | 22.92±4.16 | 0.89±0.07 | 0.94±0.03 | 0.16±0.05 |
| | Ultra-NeRF | 24.84±2.48 | 0.92±0.04 | 0.96±0.02 | 0.13±0.04 |
| | ImplicitVol | 23.56±3.55 | 0.91±0.06 | 0.96±0.02 | 0.15±0.05 |
| | UltraG-Ray (w/o Trans.) | 24.60±3.65 | 0.91±0.05 | 0.96±0.02 | 0.14±0.04 |
| | UltraG-Ray (SH=0) | 25.76±3.00 | 0.93±0.04 | 0.97±0.02 | 0.12±0.04 |
| | UltraG-Ray (no OPS[3]) | 25.99±2.74 | 0.93±0.03 | 0.97±0.01 | 0.12±0.03 |
| | UltraG-Ray | **26.92**±2.35 | **0.94**±0.03 | **0.97**±0.01 | **0.10**±0.03 |

As shown in Fig. 4, UltraG-Ray produces high-fidelity reconstructions in which individual muscle fibers are clearly distinguishable. Ultra-NeRF and ImplicitVol both oversmooth the generated images, suppressing high-frequency texture in the porcine muscle. ImplicitVol

---

[2]At the time of submission of this paper.

[3]Out-of-plane sampling

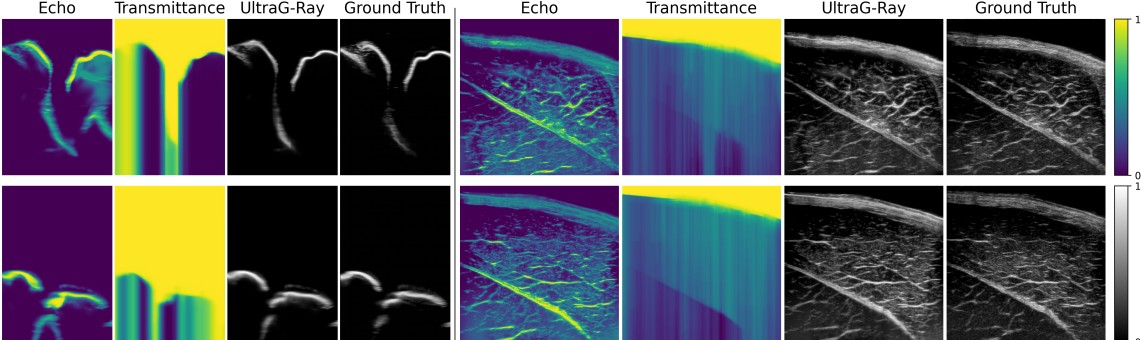

Figure 5: NVS examples from both datasets for the intermediate echo and transmittance maps, the synthesized B-mode output, and the corresponding ground truth image.

nevertheless retains finer structural detail than Ultra-NeRF, which largely recovers only the dominant fiber orientation. In contrast, UltraG-Ray resolves fine structural variations more consistently and renders them with higher contrast. The re-sliced compounding baselines preserve more local detail than the learning-based baselines, but still appear smoother cim comparison to UltraG-Ray due to interpolation during volume re-slicing. This qualitative improvement is consistent with the quantitative results in Table 1, where UltraG-Ray achieves the strongest overall performance and the gradient-based metrics reflect the enhanced structural detail. For the spine phantom, qualitative examples are shown in Fig. 3. Despite relatively high quantitative scores on the spine phantom (Table 1), Fig. 3 indicates that re-slicing baselines and methods without attenuation modeling do not recover the expected shadowing. In particular, they produce echoes underneath the spinous process, where the ground truth shows no reflections. The ablation of UltraG-Ray without transmittance exhibits the same failure, whereas the full model and Ultra-NeRF reproduce attenuation-induced shadows and better agree with the ground truth. Representative echo and transmittance maps are shown in Fig. 5, illustrating how separating transmittance from echo intensity helps capture view-dependent shadowing effects. To further assess the contribution of individual components, we conduct ablations in which we remove spherical harmonics (by setting the SH level to zero), and out-of-plane sampling (OPS), as reported in Table 1. Each ablation results in a measurable degradation in reconstruction quality on at least one dataset, indicating that the modules jointly contribute to the final performance. In addition, Appendix D reports an out-of-plane angle ablation, showing the expected monotonic performance degradation as the evaluation tilt moves further from the training distribution. Although we cap the scene at 500k Gaussians, the optimization does not always reach this limit: the porcine muscle dataset increases to the full 500k, whereas the spine phantom converges with only about 90k Gaussians. Further analysis of how the Gaussian count influences image quality is provided in App. B. At inference, this yields approximately 680 fps for the spine phantom and about 95 fps for the porcine muscle, with peak GPU memory of 0.09 GB and 0.32 GB (allocated), respectively.

## 5. Discussion & Conclusion

In this paper, we present UltraG-Ray, a novel approach to representing and synthesizing ultrasound images that combines the flexibility of 3D Gaussian fields with the fidelity of a physically grounded ray casting model. Unlike previous Gaussian-splatting approaches for ultrasound, which rely on intensity–opacity blending, our formulation directly incorporates ultrasound phenomena such as attenuation, view-dependent backscattering, and ray-aligned energy decay. These allow UltraG-Ray to capture shadowing behaviour and reflection amplitude, which is essential for realistic NVS. Our experiments on both in-silico and ex-vivo phantoms demonstrate that UltraG-Ray can synthesize realistic B-mode images with higher fidelity than state-of-the-art methods. The improvements in image quality metrics indicate that our ultrasound formation model helps resolve view-dependent ambiguities.

Despite its benefits, the method has limitations. First, it relies on a single straight-ray approximation that cannot model complex wave phenomena such as interference or multi-path propagation. Extending the model with secondary ray paths, as used in natural-image rendering, could improve physical fidelity. Second, the Gaussian field is an explicit representation. Although interpretable and easy to manipulate, it lacks inherent spatial continuity, which can leave sparsely sampled regions poorly represented. In practice, acquisition density remains a key deployment constraint. While in-plane coverage is typically sufficient, reconstruction quality depends strongly on the spacing between consecutive slices (i.e., acquisition speed) and on the amount of overlap provided by multiple sweeps. OPS partially mitigates limited coverage by providing additional constraints away from the observed planes, but it cannot fully compensate for sparse acquisitions. In the porcine muscle dataset, disabling OPS leads to a considerable performance drop, suggesting that further reducing the available information, for example, by increasing the inter slice spacing beyond the current $\approx 0.5$ mm or by relying on fewer overlapping sweeps (two in our setting), would likely degrade reconstruction stability and novel view consistency. Incorporating anatomical or statistical priors may allow further generalization to unseen tissue and improve continuity across viewpoints, which we plan to validate on in-vivo data across. In addition, we observe that the method is strongest at interpolation between observed poses but struggles with extrapolation beyond the training distribution, which is expected given ultrasound's strongly view-dependent appearance. This also motivates a practical angle-selection guideline: when B-mode appearance is dominated by speckle and rapidly decorrelating fine structures, smaller tilt ranges are recommended to avoid multi-view inconsistencies and oversmoothing.

Beyond NVS, the explicit Gaussian parameterization enables simple post-processing and provides a natural interface for future extensions. Likewise, NeRF-based methods for ultrasound suggest promising applications beyond NVS, including occupancy estimation (Wysocki et al., 2025) and shape completion (Wysocki et al., 2026). Building on top of our proposed representation, pruning low contributing Gaussians can offer a straightforward mechanism for perceptual smoothing, while associating subsets of Gaussians with anatomical labels could support semantic and structure-aware rendering and editing. Overall, UltraG-Ray demonstrates that combining an explicit 3D Gaussian scene representation with a physically grounded ultrasound formation model improves the fidelity of view-dependent effects in ultrasound NVS, and provides a promising basis for more structured and controllable ultrasound scene models.

## Acknowledgments

We thank ImFusion GmbH for providing access to their software, which supported the data acquisition and processing in this work.

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

## Appendix A. Training and implementation details

**Optimization.**   We train UltraG-Ray using the Adam optimizer with parameter-specific learning rates. The learning rate for 3D Gaussian means is set to $1 \times 10^{-4}$, for Gaussian scales and quaternions to $5 \times 10^{-3}$, and for transmittances to $5 \times 10^{-4}$. For the spherical harmonic (SH) coefficients, we use $5 \times 10^{-3}$ for the zero-order band and $1 \times 10^{-5}$ for the first-order band. All Gaussians are initialized with a transmittance of 0.99 and an isotropic scale of 0.5 mm. The SH expansion starts at degree 0 and is increased to degree 1 after 1000 iterations, which progressively introduces higher-frequency detail in a stable manner. A learning-rate scheduler decays all rates smoothly to 10% of their initial value by the end of training. We use a batch size of 8. We found that this parameterization yields the most stable optimization and the highest reconstruction fidelity across datasets.

**Refinement Strategy**   For the ultrasound optimization strategy, we employ an importance-driven Gaussian refinement strategy. Refinement is triggered every $r = 2500$ iterations, with importance scores accumulated over the same interval. Refinement begins after 1k epochs and terminates at 20k epochs, while training continues until 30k epochs. Gaussians whose scales fall below $s_{\min} = 5 \times 10^{-5}$ or exceed $s_{\max} = 5\,\mathrm{mm}$ are pruned, and the total number of Gaussians is capped at $N_{\max} = 500k$ to balance reconstruction fidelity and computational efficiency.

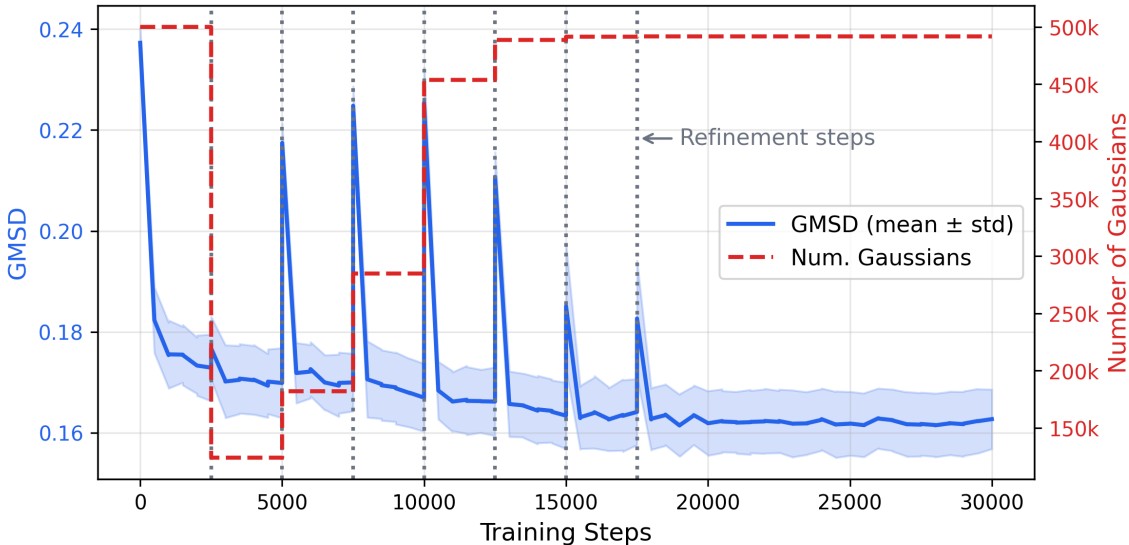

Figure 6: Evolution of GMSD scores of novel views for the porcine muscle dataset and number of Gaussians over training steps.

**Optimization Dynamics**  In order to demonstrate our optimization strategy, we evaluate our datasets every 500 iterations and capture statistics of the Gaussian parameters. Fig. 6 clearly illustrates the refinement steps every 2500 iterations. Due to random initialization, a considerable number of Gaussians do not contribute meaningfully to reconstruction. Scale regularization continuously shrinks those Gaussians, and most are subsequently pruned during the first refinement cycle due to their small scales. This can be observed as a sharp reduction in Gaussian count after the first refinement step. In subsequent cycles, the Gaussian count increases again due to Gaussians being split/duplicated in order to model increasingly fine detail. We also observe a temporary degradation in metrics after applying refinement, since duplication/splitting adds Gaussians that require optimization before contributing in a positive manner. For this reason, we disable refinement for the last 10k Gaussians.

## Appendix B. Impact of Gaussian Count on Image Quality

We evaluate how the number of Gaussians affects image quality by performing an ablation on the porcine dataset. As shown in Table 2, reducing the number of Gaussians results in only small changes in the image quality metrics. We only start seeing a noticeable reduction in these metrics below 10k Gaussians, demonstrating our method's ability to adapt to varying levels of geometric detail. This allows us to sacrifice fine detail preservation in favor of faster training and inference speeds.

These results highlight the flaws of standard evaluation metrics when applied to ultrasound imaging. Fig. 4 reveals a significant disparity in detail. The variant with 10k Gaussians manages to capture the overall anatomical structure but fails to model speckle

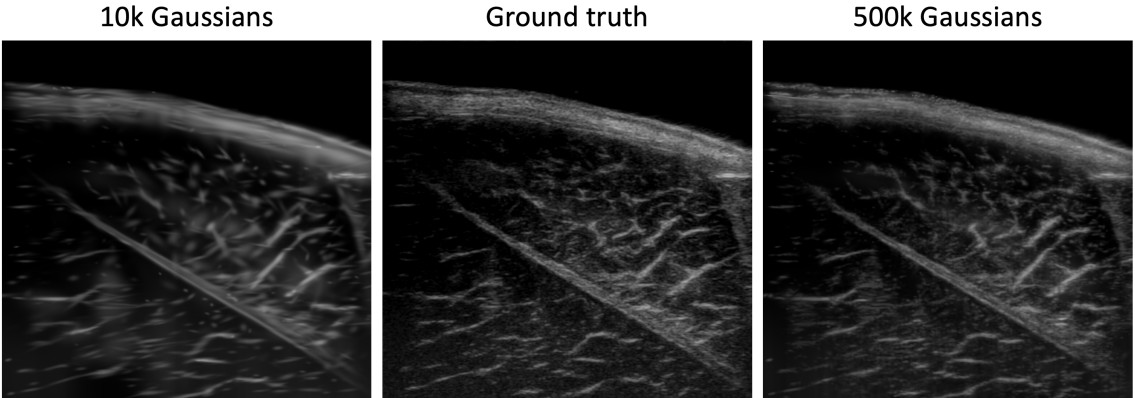

Figure 7: Comparison of novel views with differing number of max. Gaussians in the porcine muscle dataset. The metrics are comparable with Table 2.

Table 2: Comparison of image quality metrics for different maximum numbers of Gaussians.

| Dataset | #Gaussians | PSNR ↑ | MS-SSIM ↑ | GMS ↑ | GMSD ↓ |
|---|---|---|---|---|---|
| Porcine Muscle | 500k | $25.49_{\pm 0.60}$ | $0.77_{\pm 0.02}$ | $0.88_{\pm 0.01}$ | $0.16_{\pm 0.01}$ |
| | 300k | $25.39_{\pm 0.62}$ | $0.77_{\pm 0.02}$ | $0.87_{\pm 0.01}$ | $0.17_{\pm 0.01}$ |
| | 100k | $25.46_{\pm 0.66}$ | $0.77_{\pm 0.02}$ | $0.87_{\pm 0.01}$ | $0.17_{\pm 0.01}$ |
| | 10k | $25.39_{\pm 0.54}$ | $0.75_{\pm 0.01}$ | $0.85_{\pm 0.01}$ | $0.18_{\pm 0.00}$ |

and other fine textures. Even so, the variant with 10k Gaussians manages to achieve quality metrics similar to higher Gaussian counts. Here, GMS and GMSD manage to detect the degradation, even if it is larger than the score difference might suggest.

## Appendix C. Baseline Implementation Details

We trained Ultra-NeRF (Wysocki et al., 2024) and ImplicitVol (Yeung et al., 2021) using the authors' official implementations and default configuration files with no hyperparameter tuning. For Ultra-NeRF, we reused the provided dataloader after adapting our pose transformation to match the imaging axis convention, with the ray direction along y instead of z. For ImplicitVol, we provided ground truth poses for all frames and disabled pose learning to avoid pose appearance ambiguity. We additionally adapted the dataloader to our file format. Both baselines were trained until convergence on the training data using the original optimizer and learning rate schedule. All methods used the same input preprocessing and the same train-test split.

## Appendix D. Angle Generalization Ablation

We analyze how performance degrades when evaluating UltraG-Ray at probe tilt angles that progressively depart from the training distribution. The model is trained on the same two in-plane sweeps as in the main experiments ($0°$ and $+3°$) and evaluated on increasingly out-of-plane tilts on the porcine muscle dataset. Table 3 reports results for $-3°$ (identical to the main evaluation) and additional test angles further away from the training range. As expected, reconstruction quality decreases monotonically with increasing angular deviation, reflecting the growing appearance changes and limited overlap between the training views and the queried out-of-plane slices.

Table 3: Out-of-plane generalization on the porcine muscle dataset. We evaluate UltraG-Ray at increasingly out-of-plane probe tilt angles on the same training dataset (two sweeps at $0°$ and $+3°$. $-3°$ represents the same values as in Table 1, and the remaining evaluations are further tilts. Best values are in bold. Arrows indicate whether higher ($\uparrow$) or lower ($\downarrow$) values correspond to better performance.

| Eval. angle | PSNR $\uparrow$ | MS-SSIM $\uparrow$ | GMS $\uparrow$ | GMSD $\downarrow$ |
|---|---|---|---|---|
| $-3°$ | $\mathbf{25.42}_{\pm 0.59}$ | $\mathbf{0.77}_{\pm 0.02}$ | $\mathbf{0.87}_{\pm 0.01}$ | $\mathbf{0.16}_{\pm 0.01}$ |
| $-5°$ | $23.96_{\pm 0.54}$ | $0.70_{\pm 0.04}$ | $0.86_{\pm 0.01}$ | $0.18_{\pm 0.01}$ |
| $-7°$ | $22.77_{\pm 0.54}$ | $0.63_{\pm 0.03}$ | $0.85_{\pm 0.01}$ | $0.19_{\pm 0.00}$ |
| $-10°$ | $21.59_{\pm 0.35}$ | $0.55_{\pm 0.03}$ | $0.83_{\pm 0.00}$ | $0.20_{\pm 0.00}$ |

