# OpenReview forum: "UltraG-Ray: Physics-Based Gaussian Ray Casting for Novel Ultrasound View Synthesis"
_MIDL.io/2026/Conference — MIDL 2026 Poster_

### Official Review · Reviewer_qBmC · 2025-12-29

**Confidence:** 3
**Preliminary Rating:** 4
**Final Rating:** 5

**Summary:**

The study proposes UltraG-Ray, a physics-based ultrasound novel view synthesis method that replaces standard radiance-based Gaussian splatting.

The key idea is to decouple attenuation and echo formation by modeling the B-mode intensity as $B(p) = T(p) \cdot E(p)$, where attenuation is accumulated multiplicatively along probe-to-tissue rays and echo intensity is evaluated locally using anisotropic Gaussians with view-dependent backscattering.

Experiments on an in-silico spine phantom and an ex-vivo porcine muscle dataset show consistent improvements over an ultrasound NeRF baseline, particularly in preserving speckle and fine structural detail. Compared to the baseline, the proposed method yields approximately 15 percent improvement in MS-SSIM and better gradient-based metrics. Ablation studies indicate that the explicit transmittance modeling is a key contributor to these gains.

**Strengths:**

- Overall, the work is significant because it integrates ultrasound physics into a modern Gaussian splatting representation, enabling more realistic and interpretable novel-view synthesis, which is a significant improvement compared to existing works on ultrasound novel view synthesis with Gaussian splatting.

- The proposed method is technically sound, with detailed descriptions of both method formulation and practical implementation details, which indicates good reproducibility.

- The proposed method shows good performance both quantitatively (Table 1) and qualitatively (Fig. 3).

- The echo and transmittance components of the model, shown in Fig. 4 also make sense, which reflects the interpretability of the model.

**Weaknesses:**

- Section 3 is technically solid. However, it's loaded with graphics jargons that are not friendly to broader ultrasound or medical imaging audience that MIDL serves. It's beneficial to explain terms and concepts (especially in 3.2), such as "ray casting", "Beer–Lambert law", "Gaussian integrals", "spherical harmonics", "3-point Gauss-Legendre quadrature", "canonical space", etc. Plus, it would be beneficial to explain what direction the wave travels in $T(p)$ and $E(p)$ in Eq. 5.

- More ablation studies are suggested. For example:
    - Echo view-dependence. The authors may replace $I_i(d)$ (degree-1 SH) with a constant scalar $I_i$.
    - Soft coverage $\gamma (p)$ ablation.
    - 3D Mahalanobis echo weighting vs 2D projection weighting.

- The manuscript may benefit from more discussion on real-world application scenarios. For example, the intended offline usage given its training time and data requirements (i.e. how densely a region needs to be sampled). No new experiments are expected, but some discussion should be helpful.

**Detailed Comments:**

Besides what's mentioned in Weaknesses, I think the NeRF baseline may not reflect the SOTA of NVS problem in ultrasound, given the blurriness of the result in Fig. 3. I understand that there are no publicly available source codes for UltraGS or UltraGauss, but I have three more questions: 1) how were the hyperparameters of Ultra-NeRF tuned for the datasets? 2) Are there any other SOTA methods for NVS problem, such as generative models? 3) Is it possible to compare the result figures to those in UltraGS and UltraGauss qualitatively?

**Justification Of Final Rating:**

The authors’ rebuttal addressed most of my concerns and clarified the remaining issues, including why some of the requested baselines and ablation studies were not feasible, either in general or within the rebuttal time constraints. I believe the revisions improve the quality of the work, and I therefore raise my rating to 5.

**Justification Of The Preliminary Rating:**

The proposed method is innovative: it incorporates ultrasound physics (i.e. transmittance and echo) into Gaussian splatting modeling for ultrasound novel view synthesis. The proposed method is technically sound and the results are good both quantitatively and qualitatively. However, stronger baselines and more ablation studies are expected. The writing style of Section 3 can be improved by avoiding or explaining graphics jargon that is not familiar to the ultrasound community.

**Questions To Address In The Rebuttal:**

See Weaknesses and Detailed Comments.

---

> ### Author Response · Authors · 2026-01-23
>
> We thank the reviewer for the careful reading and constructive feedback. We appreciate the positive assessment of the paper’s significance in integrating ultrasound physics into a modern 3D Gaussian representation, and we are pleased that the formulation and implementation details were found technically sound and reproducible. In particular, we value the recognition that explicit transmittance modeling is a key contributor and that separating echo and transmittance components enhances interpretability. Below, we address the reviewer’s concerns point by point and refer to the clarifications and discussion improvements incorporated in the revised manuscript.
>
> ### 1. Technical readability
>
> We revised Section 3 to be more accessible to a broader ultrasound/medical imaging audience by reducing graphics jargon and adding brief explanations of key terms (especially in Sec. 3.2). For Eq. (5), we now state explicitly that the ray direction is always perpendicular to the probe surface (along the probe normal). While $T$ is evaluated along this direction (from the probe into the tissue), $E$ is evaluated at the point $\mathbf{p}$ within the tissue. While we cannot expand every concept in full detail due to space constraints, we believe these additions improve readability without changing the technical content.
>
> ### 2. Ablation studies
>
> We agree and have added additional ablations and robustness analyses (Table 1 and Appendix D). Concretely, we test echo view-dependence by setting SH=0 (constant scalar per Gaussian), removing out-of-plane sampling, removing attenuation, and evaluating angle generalization by predicting sweeps at progressively larger tilt deviations from the training distribution. The remaining suggestions are unfortunately out of scope for this revision. A soft-coverage-factor ablation is not straightforward to isolate without introducing an alternative formulation. Additionally, comparing 3D Mahalanobis echo weighting to a 2D projection weighting would require implementing explicit lateral projection of each Gaussian onto the imaging plane to compute 2D contributions, which is a non-trivial extension beyond the rebuttal scope. We hope these additions and clarifications address the intent of the requested ablations and make clear which components drive the observed improvements.
>
> ### 3. Discussion on usability
>
> We expanded the Discussion to better situate UltraG-Ray in realistic application settings. In particular, we clarify that the current method is intended for offline, scan-specific optimization, and we discuss practical data requirements in terms of pose coverage, sweep overlap, and sampling density needed to obtain stable reconstructions. Based on the added out-of-plane evaluation, we also comment on how performance degrades as viewpoints move farther from the acquisition angles, which helps quantify the limits of extrapolation.
> In addition, we strengthened the Conclusion by highlighting that the explicit Gaussian parameterization enables simple post-processing and future extensions, e.g., perceptual smoothing via pruning low-contributing Gaussians and semantic, structure-aware rendering/editing via anatomy-linked Gaussian subsets. Finally, we hope that our Gaussian splatting for ultrasound provides a compatible foundation for leveraging ongoing advances in the broader Gaussian-splatting community.
>
> ### 4. Additional baselines
>
> We investigated additional baselines and have now included one implicit reconstruction method as well as two traditional volume-based re-slicing baselines. These represent the most appropriate and reproducible comparators we could identify for our experimental setting. For generative-model baselines, we attempted to run NeRF-US. However, their released code does not fully expose the components required to reproduce the full pipeline, in particular, the diffusion model and the data used for fine-tuning this diffusion model (as noted in the repository issues). As a result, a comparison would have required training a diffusion model on our data (which is too small for adequate training), introducing substantial deviations from the original method/result that would compromise the fairness and interpretability of the comparison. Similarly, a purely qualitative comparison to UltraGS and UltraGauss would be inappropriate, since it would rely on screenshots from published figures rather than running the methods under matched conditions on our datasets.
>
> Overall, we believe these revisions substantially improve the manuscript in clarity, completeness, and practical framing. We reduced graphics-specific terminology and added short explanations where needed. We also expanded the experimental validation with additional ablations to better isolate which components impact performance. Finally, we added baselines to better position our contribution. We thank the reviewer again for the constructive feedback, which directly helped make the paper easier to understand and more complete.

---

> ### Comment · Area_Chair_nwnq · 2026-01-30
> **Please update your rating**
>
> Hello and thank you again for reviewing for MIDL !
> This is a friendly reminder to please update your rating based on author's rebuttal.
> This is really important to complete the review process and for the acceptance/rejection of papers.
> The deadline is tomorrow (February 1st 2026, 23:59 AoE).
> Thank you!

---

### Official Review · Reviewer_YjJX · 2026-01-02

**Confidence:** 4
**Preliminary Rating:** 5
**Final Rating:** 5

**Summary:**

This paper presents UltraG-Ray, a novel view synthesis (NVS) method in ultrasound imaging that combines 3D Gaussian representations with physics-based ray casting. The authors propose to encode ultrasound-specific parameters (like attenuation and reflections) into the learnable Gaussian field and subsequently apply a ray casting scheme that models acoustic wave propagation to render images. The authors evaluate their method on two new pose-annotated datasets (that they also claim to make publicly available) and demonstrate good performance over a NeRF-based baseline (Ultra-NeRF).

**Strengths:**

This paper is very well written and addresses an interesting problem. Its main strength are:
- **Well motivated**: The authors do a great job in presenting the current SOTA for NVS in ultrasound imaging, name specific shortcomings and address those within their work.
- **Clear presentation**: The paper is really well written, and clearly explains its differences to other Gaussian-based methods like UltraGS and UltraGauss. The mathematical notation is clear and the relevant hyperparamters and the training setup are well described.
- **Nice integration of physics into the method**: Separating transmittance and echo intensity is physically grounded and an ablation confirms its importance. The proposed ray casting formulation naturally captures depth-dependent attenuation unlike projection-based Gaussian methods.
- **Interesting experiments**: I generally like the experimental design, especially that the authors go beyond simply measuring PSNR and MS-SSIM values. Especially the experiment on how the Gaussian count influences the image quality provides useful practical insights.
- **Public code and new dataset**: The authors claim to make their code, as well as their datasets available upon acceptance. I think that this will be a valuable resource for others.

**Weaknesses:**

Even though this is a good paper, it has some weaknesses:
- **Limited baseline comparison**: The authors only compare their method to Ultra-NeRF. UltraGS and UltraGauss code unavailability is acknowledged, but I still think that this is a shortcoming of the paper. Comparing to some other methods (like computational sonography) could strengthen the paper.
- **Dataset limitations**: While the presented results are nice, a more comprehensive evaluation on some in-vivo data or on more anatomies and varying tissue types would have been nice to see.

**Detailed Comments:**

I have some additional comments/ questions:
- How were $\gamma_{\text{ssim}}$ and $\gamma_{\text{scale}}$ determined? How sensitive is the method to these parameters? I have the same question for the out-of-plane perturbation parameter $\delta$.

- I think that adding some training times, as well as required GPU memory could further strengthen the paper.

**Justification Of Final Rating:**

I still think this is a solid technical contribution that should be presented at MIDL. My remaining concerns have been addressed: (1) the authors added more comparisons, also to simple baselines, (2) they comment on how specific hyperparameters were determined and (3) they added some practical comments on training times and required GPU resources. I also really like the additional ablations. I can only recommend to accept this work and believe that the rebuttal further strengthened the paper.

**Justification Of The Preliminary Rating:**

Even though this paper has some shortcomings, I think that it is a solid technical contribution with clear improvements over existing methods. The proposed method is interesting, grounded in physics and really well-presented. I can only recommend to accept this paper and hope to see it presented at MIDL.

**Questions To Address In The Rebuttal:**

I'd like the authors to address my comments in the "Detailed Comments" section.

---

> ### Author Response · Authors · 2026-01-23
>
> Thank you for the detailed review and for the thoughtful assessment of UltraG-Ray. We appreciate the positive evaluation of the paper’s motivation, clarity, and technical rigor, as well as the recognition that our formulation is physically grounded and reflects key aspects of ultrasound image formation. We are also grateful for the constructive suggestions regarding broader baseline comparisons, dataset scope, and additional implementation details. In the revised manuscript, we have incorporated the requested clarifications, added practical runtime and GPU memory statistics to strengthen reproducibility, and expanded the discussion to better contextualize limitations and applicability. Below, we respond to your comments point by point:
>
> ### 1. Limited baseline comparison
>
> We agree that a broader baseline comparison would strengthen the paper. In the revision, we therefore added additional baselines beyond Ultra-NeRF, including one implicit reconstruction baseline (ImplicitVol) and two traditional volume-based compounding and re-slicing baselines (median and maximum compounding followed by re-slicing). Unfortunately, the code for computational sonography is not publicly available, making a direct comparison difficult.
>
> ### 2. Dataset limitations
>
> We agree that a more comprehensive evaluation on in-vivo data and additional anatomies would further strengthen the evidence for robustness and clinical applicability. In this submission, we focus on two controlled, pose-annotated datasets that allow systematic ablations and a clear assessment of view-dependent effects under reproducible conditions. We now outline in-vivo validation as a key next step in the revised manuscript. In parallel, we are collecting additional datasets with broader anatomical and tissue diversity and intend to include them in a planned journal extension.
>
> ### 3. Determination of parameters
>
> We empirically determined $\lambda_{\text{ssim}}$ and $\lambda_{\text{scale}}$ to work well across both datasets. For $\lambda_{\text{ssim}}$, we did not want to emphasize either $L_1$ or $L_{\text{SSIM}}$, so we weighted them equally. For $\lambda_{\text{scale}}$, we chose a value that shrinks Gaussians to enable pruning, but does not collapse the Gaussian set so quickly that it inhibits optimization. Changing this parameter would likely also require adjusting learning rates for the other Gaussian parameters and updating the pruning strategy. Both parameters can be tuned further to potentially find a better overall configuration; however, the reported setting performed best in our experiments. Regarding the out-of-plane perturbation parameter, we now include a new ablation that removes it. This ablation shows that the method overfits to the training views, resulting in reduced evaluation metrics for novel views. We derived the perturbation magnitude from the pose spacing in our datasets to ensure sufficiently diverse training rays, while remaining within realistic limits based on the elevational beam width of clinical transducers (see Section 3.4).
>
> ### 4. Training times and memory consumption
>
> We agree that training times play a significant role in whether our method would be applicable in a clinical setting. Section 4 now reports training and inference runtime as well as GPU memory usage for training and inference. Training times vary substantially between the two datasets, since the spine phantom is less complex and leads the method to converge to fewer Gaussians. In general, training time for Gaussian splatting depends strongly on the Gaussian count, since frames with more Gaussians take longer to render.
>
> Across the revision, we expanded the set of baselines (ImplicitVol, resliced median/max compounding), clarified the rationale for key hyperparameters, added an ablation for out-of-plane sampling, and reported inference and training statistics to better contextualize practical applicability. We again thank the reviewer for the careful and constructive feedback.

---

> > ### Comment · Reviewer_YjJX · 2026-01-26
> >
> > I'd like to thank the authors for this strong rebuttal. My concerns have properly been addressed and I have no remaining questions or comments. I really like the additional baselines and comments (on relevant hyperparameters and required resources), as well as the extended ablation of the method.

---

### Official Review · Reviewer_Hr1N · 2026-01-06

**Confidence:** 5
**Preliminary Rating:** 2
**Final Rating:** 3

**Summary:**

The work presents a method for using ultrasound-specific ray tracing to improve gaussian splatting of ultrasound scenes, extending UltraGauss with a transmittance optimization to model view-dependent shadowing. The method is evaluated by optimizing gaussian representations of two ultrasound recordings (one simulated, one ex-vivo), and generating novel views of these scenes.

**Strengths:**

The work has clear novelty, aiming to show that optimizing transmittance of each gaussian (which UltraGauss does not model) can meaningfully improve the representation quality.
The results that are shown look promising.

**Weaknesses:**

There is no hold-out set; hyper parameters were optimized on the same scenes results are reported on.
The data description is missing important details.
The experiments are not well described.
Important details are missing to understand what the results are showing.

**Detailed Comments:**

There is no hold-out set. The authors optimized their hyper-parameters for two specific scenes, meaning part of the performance gap between the method with and without transmittance enabled may be explained by hyperparameter overfitting to the two scenes that are evaluated.

The presented baseline is questionable. The method is only compared to UltraNerf (which seems to fail entirely in this task, judging by Figure 3) and to an ablated version of the presented method in which the transmittance optimization is disabled (of which no qualitative examples are included). There are also no simple baselines (such as voxelwise mean and/or voxelwise maximum of the input b-modes after the same pre-processing used for the method) to give context to the added benefit of using gaussian splatting to represent the scene. How does the method compare to a voxelwise fit of the same data, with or without transmittance optimization, and with or without directional intensities modeled as spherical harmonics?

The description of the data is very sparse. It is not clear how many frames are contained in the sweeps, and what the distance is between different acquisition angles. It is also not clear what the distance is between the generated novel views and their nearest neighbour in the training set (both absolutely and conceptually).

The "echo" maps being visualized using the viridis colormap while the b-modes are in grayscale makes it difficult to compare the two and appreciate their differences.

The method requires sweeps from multiple angles, but it is unclear how many the authors used for optimizing their gaussian representations, and how many (and which ones) were held out for novel view synthesis evaluation. How does the method perform when only the first two or three angles are used, and (how much) does performance degrade when evaluating on angles progressively further away from the angles used for optimization?

**Justification Of Final Rating:**

The inclusion of a more complete experiment description in the revised manuscript is very helpful. Unfortunately, the rebuttal does not assuage my concerns regarding generalization and applicability. As shown in the new appendix, tilting just 2 degrees further already degrades performance to a PSNR of less than 24, putting the method closer to the (clearly unusable) Ultra-NeRF results. This calls into question the value of the method, as there is only a very narrow region in which the method outperforms naive compounding and reslicing. The authors have unfortunately not included the compounding baseline results for the experiments in Table 3, making it unclear where the crossover point is (already before 5 degrees? Or between 5 and 7?). The main claim of the paper, i.e. modeling transmittance when using gaussian splatting for novel view synthesis in ultrasound improves reconstruction quality, is also not investigated in this region.

**Justification Of The Preliminary Rating:**

The work demonstrates a novel approach, but many important details are missing from the paper, making it difficult to judge the value of the presented results. Additionally, the paper is missing baselines to put the method performance into perspective.

**Questions To Address In The Rebuttal:**

see: detailed comments

---

> ### Author Response · Authors · 2026-01-23
>
> Thank you for the detailed feedback and for acknowledging the novelty of optimizing per-Gaussian transmittance to model view-dependent shadowing. We agree that the experimental setup can be improved. We have revised the paper accordingly and would like to answer your raised concerns in detail below.
>
> ### 1. Evaluation protocol and overfitting concerns
>
> Our method is evaluated in a scene-specific optimization setting, where we fit a representation to a scene (several overlapping sweeps) and assess performance on a held-out set from the same scene. The held-out set is defined by unique poses from a sweep that differs from the training sweeps in terms of acquisition angle and acquisition positions (though the test sweep generally overlaps with the training sweeps). We emphasize that we use a single, fixed set of hyperparameters across both datasets, which represent two substantially different scenes (in-silico spine with strong specular reflectors vs. ex-vivo porcine muscle with heterogeneous scattering). The fact that the same training parameters are used unchanged across these scenes reduces the risk that the observed gains are driven by dataset-specific tuning. In this submission, we focus on two controlled, pose-annotated datasets that allow systematic ablations and a clear assessment of view-dependent effects under reproducible conditions. Nevertheless, we agree that broader validation on additional datasets is important, and we plan to expand the evaluation in future work toward a journal submission.
>
> ### 2. Additional (simple) baselines
>
> Thanks for pointing us to these traditional baselines. In addition to Ultra-NeRF, we now include both an implicit learning-based baseline (ImplicitVol) and traditional, non-learning-based volume re-slicing. The latter are computed via classic maximum- and median-based compounding into 3D volumes, followed by re-slicing with linear interpolation. We demonstrate improvements over these new baselines both qualitatively and quantitatively (compare Table 1 and Figures 3 and 4). We also include a qualitative sample in Figure 3, demonstrating the improvements achieved by including transmittance optimization.
>
> ### 3. Dataset and split description
>
> We agree that the dataset description was insufficient. We therefore expanded the dataset paragraph with the missing acquisition and split details:
>
> - **Porcine muscle:** three sweeps acquired at -3°, 0°, and +3° (about 100–120 frames each), covering roughly 5 × 5 × 5 cm³. We train on 0° and +3° and evaluate on -3°. Results on additional held-out sweeps at -5°, -7°, and -10° are reported in Appendix D.
> - **Spine phantom:** six sweeps acquired at -20°, -10°, 0°, +10°, +20°, plus evaluation on +15°, approximately 350–400 frames per sweep, covering a volume of roughly 13 × 5 × 9 cm³.
>
> ### 4. Visualization: Echo map
>
> We understand the concern. Showing the echo map with a viridis colormap while B-mode images are in grayscale can make side-by-side comparison less direct. However, our intent was to avoid misinterpretation, since the echo map, like the transmittance map, is an intermediate result and not a B-mode image by itself. Only their combination yields the final B-mode output. Using a distinct colormap helps clearly separate intermediate maps from the grayscale B-mode result and reduces the risk that intermediate grayscale images are perceived as additional B-mode reconstructions.
>
> ### 5. Angle generalization (out-of-plane evaluation) and ablations
>
> To directly quantify robustness as the viewpoint departs from the training distribution, we added an out-of-plane angle generalization study in which the model is trained on two sweeps and evaluated on progressively more distant tilt angles (Appendix D). As expected, performance decreases monotonically with increasing angular deviation. In addition, we include targeted ablations that disable directional intensity modeling (SH=0) and out-of-plane sampling (OPS) to isolate their impact on reconstruction quality (Table 1).
>
> We thank the reviewer again for the constructive comments. We believe the revised manuscript improves clarity and places UltraG-Ray in a stronger context by comparing against both learning-based methods and traditional volume re-slicing baselines. By demonstrating qualitative and quantitative improvements over these baselines, we aim to clearly substantiate the impact of the proposed approach. The added ablation studies further isolate and validate the contributions of transmittance optimization, directional intensity modeling, and out-of-plane sampling. We hope these revisions address the main concerns and make the contribution and added benefits of the method straightforward to assess.

---

### Author Rebuttal · Authors · 2026-01-23

**Rebuttal:**

We thank the reviewers for their careful reading of our submission and for the constructive feedback. We have addressed each point in a detailed, point-by-point response below each review. The revised manuscript is attached, with changes highlighted in blue. For new figures, the corresponding caption text is highlighted in blue.

**Supporting Material:**

/attachment/690f75d2f32a9812a984de3034cd805761b0606a.pdf

---

### Meta-Review · Area_Chair_nwnq · 2026-02-06

**Recommendation:** Accept (Poster)
**Confidence:** 4

**Metareview:**

The reviewers have all appreciated the novelty of the proposed method, which they found technically sound and well-justified in terms of physics. They also agree that the results are promising, especially since the authors have added more baselines and ablations after the rebuttal. The authors have also improved the technical readability of the methods, and have added needed dataset information.

In terms of limitation, the rebuttal highlighted a generalisation problem away from the angles and areas seen at training. This problem has been acknowledged and its now discussed in the conclusion along with the offline nature of UltraG-Ray and future extensions to new in-vivo datasets.

Overall, all concerns have been addressed or discussed, and I can only recommend a well-deserved acceptance for this paper. However, I leave the oral/poster decision to the PC, because this work requires strong computer graphics knowledge to be understood, which might not be easily accessible to the MIDL community.

---

### Decision · Program_Chairs · 2026-02-13

Accept (Poster)